# Innate Immunity, Inflammation, and Intervention in HBV Infection

**DOI:** 10.3390/v14102275

**Published:** 2022-10-17

**Authors:** Ge Yang, Pin Wan, Yaru Zhang, Qiaoru Tan, Muhammad Suhaib Qudus, Zhaoyang Yue, Wei Luo, Wen Zhang, Jianhua Ouyang, Yongkui Li, Jianguo Wu

**Affiliations:** 1Foshan Institute of Medical Microbiology, Foshan 528315, China; 2Guangdong Provincial Key Laboratory of Virology, Institute of Medical Microbiology, Jinan University, Guangzhou 510632, China; 3State Key Laboratory of Virology, College of Life Sciences, Wuhan University, Wuhan 430072, China; 4Clinical Research Institute, The First People’s Hospital, Foshan 528000, China; 5Guangdong Longfan Biological Science and Technology, Foshan 528315, China

**Keywords:** hepatitis B virus, innate immunity, inflammation, signal pathway, therapeutic strategy

## Abstract

Hepatitis B virus (HBV) infection is still one of the most dangerous viral illnesses. HBV infects around 257 million individuals worldwide. Hepatitis B in many individuals ultimately develops hepatocellular carcinoma (HCC), which is the sixth most common cancer and the third leading cause of cancer-related deaths worldwide. The innate immunity acts as the first line of defense against HBV infection through activating antiviral genes. Along with the immune responses, pro-inflammatory cytokines are triggered to enhance the antiviral responses, but this may result in acute or chronic liver inflammation, especially when the clearance of virus is unsuccessful. To a degree, the host innate immune and inflammatory responses dominate the HBV infection and liver pathogenesis. Thus, it is crucial to figure out the signaling pathways involved in the activation of antiviral factors and inflammatory cytokines. Here, we review the interplay between HBV and the signal pathways that mediates innate immune responses and inflammation. In addition, we summarize current therapeutic strategies for HBV infection via modulating innate immunity or inflammation. Characterizing the mechanisms that underlie these HBV-host interplays might provide new approaches for the cure of chronic HBV infection.

## 1. Introduction

Hepatitis is a liver inflammatory response which can be repaired itself under certain conditions, but it can also progress to liver fibrosis, cirrhosis, and liver cancer [1]. The major cause of hepatitis is the hepatitis viruses. Five major hepatitis viruses have been identified—hepatitis A virus (HAV), hepatitis B virus (HBV), hepatitis C virus (HCV), hepatitis D virus (HDV), and hepatitis E virus (HEV)—with HBV being the most infected and dangerous, causing hepatitis B and affecting 257 million people worldwide [2]. HBV is highly contagious, spreading through mucous membranes or incomplete skin exposed to infected blood or other body fluids (saliva, semen, and vaginal fluid) [3]. Currently, inoculating hepatitis B vaccine is the safest and most effective measure to prevent hepatitis B. Statistics show that 80–95% of the population has an immune response after the whole course of hepatitis B vaccine inoculation, and its protective effect can span more than 20 years. However, the level of immune response to hepatitis B vaccine is different among different individuals, as 5~10% of healthy people cannot produce protective antibody effectively even after the whole course of Hepatitis B vaccine immunization [3,4]. HBV has a genome of around 3.2 kb consisting of a complete DNA strand and a partial DNA strand [5]. HBV is a member of the Hepadnaviridae family, which contains structural proteins (a reverse transcriptase attached to genome DNA, a core protein, and three envelope proteins) and non-structural proteins, and its infection depends on the entrance receptor, sodium taurocholate co-transporting polypeptide (NTCP) [6,7]. The replication of HBV is broadly divided into the following six stages. (1) Envelope virus adsorption: proteins expressed by HBV, such as the pre-S1 region of HBV envelope protein L, can bind to NTCP, thus facilitating the entry of HBV into cells [7]. (2) Nucleocapsid and relaxed circular DNA (rcDNA) nucleocapsid: the viral envelope is removed and the nucleocapsid containing rcDNA is released into the cytoplasm [8]. (3) Covalently closed circular (cccDNA) repair: because cccDNA is the transcription template of viral RNA, it is the first step to convert rc-DNA into cccDNA after virus infection [9]. (4) cccDNA transcription and translation: with the help of RNA polymerase II, cccDNA is transcribed as a template into viral RNAs, including prc-C mRNA/pgRNA and three subgenomic RNAs (2.4 kb, 2.1 KB, 0.7 KB) [9]. (5) pgRNA packaging and reverse transcription: packaging reverse transcriptase and pgRNA-specific into the newly formed capsid is a critical step in hepadnavirus replication [9]. (6) Virion assembly: the nucleocapsid containing rcDNA is transported to the endoplasmic reticulum (ER), and the virion is formed by coating the ER cavity [8].

Thus, a mature HBV particle contains an about 3.2-kb partially double-stranded rcDNA genome, which consists of a complete minus strand and a partially synthesized plus strand [8]. The intracellular cccDNA can generate from the repairmen of viral genome rcDNA in the nucleocapsid of newly infected virus. Besides, the cccDNA can also synthesized from pre-genomic RNA (pgRNA) which is conducted by the reverse transcriptase (the viral P protein) and completed with generation of a complementary strand [8].

The fate of HBV infection is determined by a complicated series of events that occur between this virus and the host immune system [10]. HBV infection stimulates quick immune responses, which usually appears as an acute self-restriction infection and brings lifetime immunity in adults; however, in newborns and children, the infection may become chronic with lifetime virus persistence [11]. During the early phase (also called the immune tolerant phase) of CHB (chronic hepatitis B) infection, due to the lack of host immune response, liver histology and alanine aminotransferase (ALT) levels are usually normal [12]. Most patients in this phase have minimal liver injury, and prognosis is favorable during follow-up of up to 10 years [12]. Thus, if the patients can receive effective antiviral therapy during this phage, they may avoid suffering from progression to hepatitis, cirrhosis, or HCC. It is widely assumed that adaptive immune response plays a significant role in HBV infection clearance [13]. On the other hand, the function of innate immunity in HBV infection does not appear to be well known, which maybe lead to the fact that no effective approaches have been developed to treat the early subclinical stage of HBV infection [14,15]. As experimental models for research advance greatly, it has progressively been realized that the role of temporal and spatial immunological alterations as well as innate immune response is important for the process of HBV infection.

## 2. Activation of Innate Immune and Inflammatory Response by Hepatitis B Virus

The viral components can quickly stimulate host cell pattern recognition receptor (PRRs) when virus infects cells. As a result, the expression of type I interferon and inflammatory cytokines is activated. The up-regulation of pro-inflammatory cytokines and interferon-stimulated gene (ISG) expression in infected cells a the common feature of different virus infections [16,17]. This usually leads to an innate antiviral response that controls infection until it is resolved by an adaptive immune response. However, some viruses, such as influenza viruses, cause cell death due to their cytopathic effect (CPE) [16].

HBV was formerly thought to be a “stealth virus” capable of establishing chronic infection in hepatocytes by avoiding the host innate defense system, and infection-induced cell death and CPE were rarely observed in HBV infection/replication cell models [18]. HBV infections are asymptomatic during the early stage partly because of limited induction of pro-inflammatory cytokines. In a chimpanzee infection model, the level of ISG expression was not altered in the liver by the HBV infection [19]. Meanwhile, during acute HBV infection, numerous cytokines, including IFN-α, tumor necrosis factor-α (TNF-α), and IL-15 were mildly activated in blood samples of HBV patients [20]. Although accumulated evidence has revealed that HBV is recognized by innate immunity as shown inFigure 1, HBV could only induce pro-inflammatory cytokines to a very limited extent. However, the fundamental issue to be resolved is the activation of HBV by PRRs.

### 2.1. Toll-like Receptor Pathway Stimulation in HBV Infection

Toll-like receptors (TLRs) are an important component of innate immunity, which recognize pathogen-associated molecular patterns (PAMPs) such as lipopolysaccharide (LPS), double-stranded RNA (dsRNA), and single-stranded RNA (ssRNA) to induce intracellular immune response [21]. Accumulated studies have verified the activation of the TLR signaling pathway under the stimulation of HBV and its viral proteins [22,23,24,25].

B cells could be activated by HBV through the TLR2-mediated pathway, possibly related to antiviral responses in patients with chronic hepatitis B (CHB) [22]. Additionally, when incubated with M2 macrophages, the HBV core protein accelerates the release of pro-inflammatory cytokines IL-6 and TNF-α via the TLR2 pathway [23]. Hepatitis B e antigen (HBeAg) significantly elevates TLR4 expression in monocytes of CHB patients, which might control the immunotolerance caused by regulatory T cells. HBV x protein (HBx) encoded by the X gene is important for initiating and maintaining HBV replication and can bind to cccDNA to activate the transcription of viral promoters [24]. TLR4 expression was found to be accelerated by transfected HBx in immortalized proximal tubule epithelial cells, dysregulating the production of IL-6, IL-4, and so on [24]. Specifically, the overexpression of HBx in hepatic and hepatoma cell lines activates TLR4 downstream signaling components such as myeloid differentiation primary response 88 (MyD88), IRAK1, and nuclear factor-kappaB (NF-κB), contributing to the secretion of IL-6, a major pro-inflammatory cytokine [25]. It has been discovered that the overexpression of HBx can stimulate the promoter activity of major vault protein (MVP) to raise its expression, ending up inducing type-I IFN production in hepatocytes [26,27]. MVP interacts with MyD88 to promote the type-I IFN release, so it can be concluded that HBx activates TLR-mediated innate immune response indirectly. Besides, HBx can promote the protein level of DExH-box RNA helicase 9 (DHX9) which interacts with MyD88 and accelerate the type-I IFN production [28]. Stimulating MyD88 can lead to transforming growth factor-β-activated kinase 1 (TAK1) activation, and consequently causes inflammation and interferon production [21,29,30]. It is reported that HBx can activate TAK1 to induce NF-κB and promote the expression of the CXC Chemokine IP-10 [31]. The oligomerization of tumor necrosis factor receptor-associated factor 6 (TRAF6) is fundamental for its activation, and HBx has been proved to promote the oligomerization of TRAF6 to initiate its activity [32]. HBx can interact with the evolutionarily conserved signaling intermediate in Toll pathway (ECSIT), which is the partner of TRAF6, and activate NF-kB signaling pathway, resulting in the release of IL-10 [33].

### 2.2. NLR-mediated Pathway Stimulation in HBV Infection

Nucleoside binding domain and leucine-rich repeat sequence (NLR) proteins are a family of intracellular receptor that play an important role in inflammation and innate immune responses [34,35,36]. Hepatitis B core protein has been reported to induce liver inflammation through promoting LPS-mediated NLRP3 inflammasome activation and increasing the production of pro-inflammation cytokine IL-1β [37]. Furthermore, HBx promotes hepatocyte pyroptosis in a NLRP3 inflammasome-dependent manner by accelerating the release of IL-1β and IL-18 [38].

### 2.3. RLR-Mediated Pathway Stimulation in HBV Infection

Through studies on a human liver chimeric mice model, Okamoto et al. discovered that RIG-I was identified as a sensor protein for HBV pregenomic RNA, by which HBV activates NK cell-dependent innate immune responses [39]. Retinoic acid-inducible gene 1 (RIG-I)-like receptors (RLRs) sense the double-stranded RNA (dsRNA) produced by viruses and lead to activation of antiviral responses [40]. Members of the RLRs family that have been discovered so far mainly include RIG-I, melanoma differentiation-related gene 5 (MDA5), and genetics and physiology experimental gene 2 (LGP2) [41]. Lu et al. demonstrated that transfection of the HBV replicative plasmid into Huh7 cells could increase the MDA5 expression, but not RIG-I. Furthermore, HBV RNAs were found to activate MDA5 signaling [42].

### 2.4. JAK/STAT Pathway Stimulation in HBV Infection

JAK/STAT consists of three components: the tyrosine kinase receptor receiving the signal, the tyrosine kinase transmitting the signal-JAK, and the transcription factor producing the effect-STAT [43,44]. As the downstream pathway of IFN receptors, the JAK/STAT signaling must respond rapidly to initiate ISG production [45]. HBx-mediated JAK/STAT activation plays an important role in stimulating innate immune response during HBV infection. Lee et al. reported that HBx could bind to JAK1 and activate its activity [46]. When transfected into hepatoma cells, HBx promotes the production of IL-6 that participates in the activation of STAT3 [47]. Additionally, another cytokine, IL-34, was also elevated after HBV infection, and this was due to the activation of STAT3 induced by HBx [48].

### 2.5. NF-κB Signal Stimulation in HBV Infection

The NF-κB signal is stimulated by different PRRs and mediates innate immune response and inflammation [49]. When activated, NF-κB enters the nucleus, binds to the target DNA, and initiates the transcription of inflammatory genes [50]. Su et al. reported that the overexpression of HBx could activate NF-κB signaling by inducing phosphorylation of IκBa [51]. HBx interacts with p65 (a component of NF-κB complex) to activate NF-κB and induce metastasis-associated protein 1 (MTA1) that plays a vital role in inflammatory responses and tumorigenesis [52]. IkappaB Kinase (IKK) is an enzyme complex that forms part of the NF-κB signaling pathway. The IKK complex is comprised of three subunits (IKKα, IKKβ, and IKKγ), the α- and β-subunits are catalytically active, whereas the γ-subunit has a regulatory function [49]. Furthermore, the overexpression of HBx can increase the activation of the NF-κB signal to induce pro-inflammatory cytokines through interacting with IKKγ, boosting IKKα expression, and regulating IKKβ activity [53,54,55,56]. In addition to directly acting on NF-κB components, HBx can also activate the NF-κB pathway in indirect ways. HBx is able to promote NF-κB activation through regulating ECSIT, VHL-binding protein (VBP1), amplification in breast cancer 1 (AIB1) or valosin-containing protein (VCP) [33,57,58,59].

## 3. Mechanism of Immunosuppression in Hepatitis B Virus Infection

Accumulated evidence reveals that HBV has established strategies to avoid the innate immune responses and facilitate its replication [60]. After the nucleocapsid is depolymerized, the rcDNA is transported into the nucleus through the nuclear pore and is further repaired into the cccDNA [61]. As the cccDNA is the transcription template of viral RNAs, the rcDNA is converted to cccDNA after infection, which indicates the successful infection of virus [62]. The cccDNA circulation and viral DNA integration result in persistent infection of HBV [63]. It appears that HBV remains essentially undetectable to the innate sensing apparatus throughout the life cycle and is difficult to remove.

Immune evasion by inhibiting RIG-I activation via N6-methyladenosine (m6A) alteration conducted by m6A writer enzymes (METTL3 and METTL14) of HBV RNAs has previously been documented [64]. During HBV infection in HepG2 cells, the phosphatase and tensin homolog (PTEN) modified by N6-m6A could suppress the innate immune response by reducing the phosphorylation of IRF3 and decreasing the production of IFN [65]. After the infection in PHHs, HBV could also attenuate RIG-I-mediated IFN signaling activation through a complex containing hexokinase [66]. HBV infection in human hepatocytes inhibits cGAS expression and its function in spite of the fact that the hepatocyte cGAS pathway is functionally active [67]. Furthermore, the lost sense of infective HBV DNA by cGAS in primary human hepatocytes (PHH) suggests that the cGAS sensing is impaired upon HBV infection [67,68]. Our previous studies have revealed that transfection of HBV plasmids can inhibit IFN/JAK/STAT signaling and its downstream antiviral responses by inducing the expression of matrix metalloproteinase 9 (MMP-9) and collagen triple helix repeat containing 1 (CTHRC1) [69,70]. Some studies have demonstrated that NF-κB activation could lead to the suppression of HBV replication in different cell lines [71,72,73]. The infection of HBV in HepG2-hNTCP cells upregulates the expression of fibronectin to inhibit the NF-κB signaling and increase the sensitivity of HBV enhancers, facilitating viral replication [74]. Overall, evidence is mounting that HBV actively negates the host’s innate immune response via different known and unknown strategies that are involved in various components of the innate immune signaling cascade as shown in Figure 2.

### 3.1. The Immunoregulatory Roles of Hepatitis B Virus e Antigen (HBeAg) and Surface Antigen (HBsAg)

Hepatitis B e antigen (HBeAg) and surface antigen (HBsAg) are secreted proteins generated during HBV replication which are used for detecting viral infection. Many studies have revealed that they have immunoregulatory functions that help the virus evade the antiviral immune responses.

TLR2 expression in peripheral blood mononuclear cells (PBMCs) and hepatocytes in HBeAg-positive CHB patients is lower than in HBeAg-negative CHB patients [75]. However, TLR2 expression cannot be altered by the HBV plasmids, which prevents HBeAg synthesis [76]. Tali et al. reported that HBeAg binds to Toll/IL-1 receptor (TIR)-containing proteins Mal and TRAM which are important adapters in the TLR2 signaling pathway and inhibits NF-κB activation and IFN-β promoter activity [77]. Additionally, a study has revealed that the HepG2 cells stably expressing HBeAg have lower mRNA levels of IFN-α and IFN-β than control cells [78]. Wang et al. also revealed that HBeAg suppresses the NF-κB activation via disrupting TRAF6-dependent K63-linked ubiquitination of the NF-κB essential modulator (NEMO), thus promoting HBV replication and sustaining persistent viral infection [79].

Plasma HBsAg levels have been reported to be linked to decreased TLR2 and TLR4 ligand-induced proinflammatory cytokine expression in CHB patients’ PBMCs [80]. Several reports revealed that the treatment of HBsAg could suppress TLR-mediated ERK, JNK, and NF-κB signaling pathways in monocytes and macrophages [81,82,83]. In murine hepatocytes, the activity of NF-κB, IRF3, and MAPKs could also be attenuated by the overexpression of HBsAg [84]. Additionally, HBV infection is able to dysregulate the function of myeloid dendritic cells through HBsAg’s action [85]. Moreover, HBsAg treatment can inhibit the IFN-α production induced by TLR9 in plasmacytoid dendritic cells (pDCs) [86,87]. The expression levels of TLR4, -8, and -9 have also been found to be reduced in peripheral DCs subtypes from individuals with persistent HBV infection [88]. In the human myeloma B-cell line RPMI 8226, HBsAg dysregulated TLR9 malfunction by inhibiting transcription factor CREB activation [89]. Deng et al. also found that HBsAg interfered the NF-κB signaling by binding to the TAK1 and TAB2 complex, leading to an attenuated immunological response [90]. In addition, a reduction of activated NK cells was found in the liver of HBsAg-expressing mice, indicating that HBV could affect immune cell activation by regulating immunosuppressive signal pathways and cytokines [91]. HBsAg, like HBeAg, disrupts the antiviral immune responses mediated by TLRs [76,77]. Moreover, Wu et al. has found that HBV particles, HBsAg, and subviral particles could almost completely block TLR-mediated innate immune responses in mouse parenchymal and nonparenchymal hepatocytes [92].

### 3.2. The regulatory Role of HBV Core Antigen (HBcAg) in Innate Immunity and Inflammation

HBV core antigen (HBcAg), encoded by spliced pgRNA, was found to block the IFN-promoter and inhibit IFN-production in murine fibroblasts by binding to the IFN-promoter as a transcriptional suppressor [93]. Furthermore, HBcAg can inhibit IFN-induced MxA protein that is a key antiviral protein kinase [94]. As a result, HBcAg is hypothesized to facilitate HBV replication by decreasing IFN and MxA protein. Ding et al. have recently reported that the transfection of HBcAg is able to promote the LPS-mediated NLRP3 inflammasome activation and induce the release of IL-1β in HepG2 cells [37]. HBcAg was also found to have suppressive effect on the expression of interferon-induced transmembrane protein 1 (IFITM1) in HepG2 cells, contributing to HBV replication [95]. It is reported that a spliced form of pgRNA encodes a new protein called HBSP, which may also downregulate MxA production [96].

### 3.3. The Negative Regulatory Activities of HBx in Innate Immune Response

Many studies have found that HBx impairs the innate immune response through modulating the PRR signal pathways, which aids HBV replication and promotes HCC progression. Retinoic acid-inducible gene I (RIG-I) can recognize HBV pgRNA and consequently activate MAVS/MDA5/TBK1/IRF3 signaling to initiate IFN transcription, starting cell innate immune response against HBV [97,98,99]. HBx promotes MAVS ubiquitination and proteasome-mediated degradation via the Lys136 site targeted by the ubiquitin E3 ligase RNF125 [100]. HBx prevents the activation of the RIG-I, melanoma differentiation-associated gene 5 (MDA5) and IFN-β promoters to facilitate HBV replication [53,100]. Moreover, HBx could interact with MAVS to affect its signal transduction [100]. HBx can activate Parkin, which suppresses the MAVS signaling by binding to the unanchored linear polyubiquitin chain accumulated on the MAVS [101]. RNA-editing enzymes adenosine deaminase acting on RNA 1(ADAR1) is a crucial factor in the modulation of endogenous RNA-mediated innate immune response [102]. HBx up-regulates ADAR1 expression to impede the transcription of MDA5 and RIG-I genes, which suppresses HBV RNA recognition in hepatocytes [103]. HBx can impair RIG-I/MDA5-mediated innate immune response by hijacking Sp110, which is a nuclear body protein critical for RIG-I and MDA5 expression [104]. Additionally, HBx could impair IFN production by interacting with TNF receptor-associated factor 3 (TRAF3), TIR-domain containing adaptor-inducing IFN-β (TRIF), TBK1, and IRF3 [53]. HBx has been found to downregulate the expression of IFN-α receptor 1 (IFNAR1) by suppressing the activity of TYK2, and thus impairing extracellular type-1 IFN-induced signaling in Chang cells [105].

In addition to the suppressive mechanisms of IFN-mediated responses, overexpression of HBx could inhibit the transcriptional level of tripartite motif containing 22 (TRIM22), known as an antiretroviral protein in vivo and in vitro [106]. Additionally, HBx may also impair immune response by inhibiting the NLR family, such as AIM2. A study has revealed that HBx not only inhibits the mRNA expression level of AIM2 by stabilizing the enhancer of zeste homolog 2 (EZH2), but also promotes AIM2 protein degradation in a proteasome-dependent manner in HBV-infected hepatocellular carcinoma (HCC) cells [107].

### 3.4. Inhibition of Innate Immunity by HBV Polymerase (Pol)

Several studies have revealed that HBV Pol interferes with IFN-mediated immune response. Overexpression of HBV polymerase (Pol) suppresses the activation of IRF and NF-κB, both of which lead to the induction of IFN-β in different cell lines [108]. HBV Pol inhibits the stimulator of interferon genes (STING) mediated IRF-3 activation in Huh7 cells [109]. By deleting the HBV Pol, HBV Pol has been proven to promote the pathogenesis and immune evasion in viral infection. A study by Wang et al. revealed that overexpression of HBV Pol could make HBV infection resistant to IFN treatment [110]. In hepatocytes, HBV Pol contributes to the suppression of IFN-β via downregulating TBK1/IKKε axis signaling [111].

In summary, several proteins encoded by HBV exert suppressive functions on cell innate immunity, helping construct and sustain HBV infection. However, the majority of the biological associations discovered above still need to be confirmed in proper and realistic infection in vivo models. The conclusions of many studies summarized above seem contradictory. For example, the effects of HBV on RLR and IFN signal transduction exhibit both stimulation and inhibition. Actually, chronic HBV infection has a complicated course with three phases identified: 1) an immune-tolerant phase with a high HBV DNA level associated with minimal liver disease; 2) an immune-active phase with a high HBV DNA level with active liver inflammation; and 3) an inactive phase with a low HBV DNA level and minimal inflammation and fibrosis on liver biopsy [112]. HBV may influence these signal pathways of innate immunity by different means in each phase of infection.

## 4. Drugs and Therapeutic Strategies Targeting Innate Immunity or Inflammation for HBV Infection

As the mechanism that HBV employed to suppress innate immunity has been uncovered, immunotherapeutic strategies for HBV infection are being developed. Approaches employing PRR agonists and based on IFN to stimulate innate immunity have been advocated as a therapy treatment for HBV infection. Simultaneously, it has been proposed that approaches that efficiently limit HBV replication and viral protein synthesis can be combined with immunoregulatory therapies in order to avoid the suppressive actions of HBV proteins on immune responses.

### 4.1. Antiviral Strategies via Targeting PRRs

It has been demonstrated that the injection of TLR3, -4, -5, -7, and -9 ligands could reduce intrahepatic HBV replication in HBV transgenic animal models in a non-cytopathic and IFN-dependent manner within 24 h [113]. Meanwhile, Wu et al. found that the agonists of TLR3, -4 were effective to suppress HBV replication in nonparenchymal liver cells by promoting innate immune responses independent of MyD88 [114]. These support the prospect of stimulating the TLR-dependent signaling to treat chronic HBV infection. Thus, TLR ligands were investigated in in vivo and in vitro assays for inhibiting HBV replication [115,116,117]. Actually, the activation of TLR2 and TLR4-mediated innate immune response can inhibit HBV and WHV replication in hepatoma cells and woodchuck hepatocytes, respectively [116]. A study by Lanford et al. revealed that the activation of TLR7 could lead to the suppression of HBV in chimpanzees [118]. TLR7 agonist, GS-9620, has been confirmed to have therapeutic efficacy in woodchuck and chimpanzee models. Quick oral treatment of GS-9620 can inhibit HBV DNA copies in serum and liver for an extended period of time, accompanied by increment of IFN-α and proinflammatory cytokines production and activation of natural killer (NK) cells [118]. Niu et al. found that GS-9620 can induce numerous antiviral factors in human PBMCs, bringing about permanent IFN-dependent HBV suppression in PHH and HepaRG cells [119].

The 5′ triphosphorylated RNA-induced activation of RIG-I and IFN-inducible dsRNA-activated protein kinase (PKR) has been proven to suppress HBV replication in in vivo and in vitro models [120,121]. The activator of RIG-I, SB9200, can upregulate IFN-α/β and ISGs activation in the blood and liver of WHV-infected woodchucks, leading to the decrease of WHV DNA and WHsAg in serum [122].

In addition to the PRR agonists, overexpression of PRR-related adaptors or RIG-I adaptors such as MyD88, MAVS, and TRIF can significantly control HBV replication in human hepatoma cells [123,124]. Our previous studies have found that several host proteins including Golgi protein 73 (GP73), MMP9, and homeobox protein A10 (HOXA10) are involved in the regulation of HBV replication, which may provide potential strategies for suppressing HBV replication or reducing infectious inflammation [69,71,125]. Therefore, effective approaches to control viral infection and alleviate infectious inflammation remain to be developed.

### 4.2. IFN-Based Inhibition of HBV Infection

Analysis of serial liver biopsies demonstrated that HBV does not cause substantial changes in the expression of type I IFNs during the early weeks of HBV infection and dissemination, and these findings were confirmed in CHB patients [19,126]. IFNs, as critical elements of the innate immune system, have been shown to limit viral replication by influencing numerous phases during HBV infection [127]. However, HBV was demonstrated to inhibit the nuclear accumulation of STAT-1 after human IFN-α treatment in a mouse model, partly explaining the lower expression of interferon-stimulated genes (ISGs) in human hepatocytes infected by HBV [128]. Long-acting polyethylene glycol (Peg)-IFN-α, on the other hand, was able to overcome the decrease of HBV-infected hepatocyte reactivity and produce prolonged increase of human ISGs [129]. Without the participation of the immune cell response, a higher antiviral action of Peg-IFN-α on human hepatocytes might cause a considerable drop in circulating HBsAg and HBeAg levels in chimeric mice. A Study by Belloni et al. also discovered that IFN-α may induce epigenetic suppression of the cccDNA minichromosome, decreasing HBV replication in humanized mice [130]. The most common side effect of IFN-α is the initial influenza-like illness characterized by slight fever and listlessness. Other common side effects include fatigue, anorexia, mild hair loss, mood swings, and so on [12]. IFN-α with antiviral activities and immunoregulatory properties is still the most effective medicine for the treatment of CHB patients in spite of low response rate and adverse effects. The administration of nucleotide analogs (NUCs), if combined with IFN-α, has been proven to be more effective especially in HBsAg clearance [131]. To optimize the antiviral effectiveness of IFNs and decrease the side effects, a deeper understanding of the antiviral mechanisms of IFN-α is required. IFN-α mediates cell-to-cell HBV resistance transmission. Li et al. revealed that IFN-α can cause viral resisting ability to be transferred from liver sinusoidal endothelial cells (LSECs) to hepatocytes through exosomes [132]. Moreover, multiple IFN effectors have been revealed to be able to suppress viral replication directly. MxA, which is an interferon-inducible protein, suppresses nuclear transport of HBV mRNA so as to attenuate the viral replication [133]. Zinc finger antiviral protein (ZAP) can inhibit HBV replication by reducing HBV RNA levels [134]. Recently, Mao et al. found that RNA Helicase DDX17, which is the cofactor of ZAP, functions as another intrinsic host antiviral factor against HBV replication via disrupting pgRNA encapsulation [135].

Beside type-I IFNs, type III IFN, IFN-λ1, 2, and 3 have been shown to have immunoregulatory effects on both innate and adaptive immune responses. IFN-λ has comparable ISGs induction activities and antiviral properties similar to IFN-α. Furthermore, IFN-λs have the potential for less side effects because they communicate through IL-10Rꞵ and IL-28Ra which are confined to certain cells, as opposed to the broadly dispersed type-I IFN receptors [136,137]. We have reported that IL-27 and IFN-λ1 are coordinated to suppress HBV replication, which may provide an effective strategy for curing chronic HBV infection [138].

## 5. Conclusions

The cell innate immunity is the host’s first line of defense against HBV infection. After infection, HBV employs evasion mechanisms to escape from the innate immune responses. Viral proteins such as HBX, HBV Pol, HBsAg, HBcAg, and HBeAg can inhibit TLRs, the JAK/STAT pathway, TBK1/IKK, and cytokines through multiple means. Experimental data have underlined the relevance of innate immunity in the regulation of HBV infection. Both in vitro and in vivo experimental models have deepened our understanding of innate immunity and inflammation in HBV infection, providing essential basis for HBV treatment. Of all the molecules mentioned above, IFN-α is the most promising therapeutic target, as IFN-α not only has a direct antiviral effect, but also enhances the response of NK cells. However, the mechanism by which HBV escapes from innate immune identification is still partly unknown. The relationship between HBV and host immunity may be better elucidated in future by employing novel experimental models or techniques which will help develop effective therapeutic strategy for HBV infection.

## Figures and Tables

**Figure 1 viruses-14-02275-f001:**
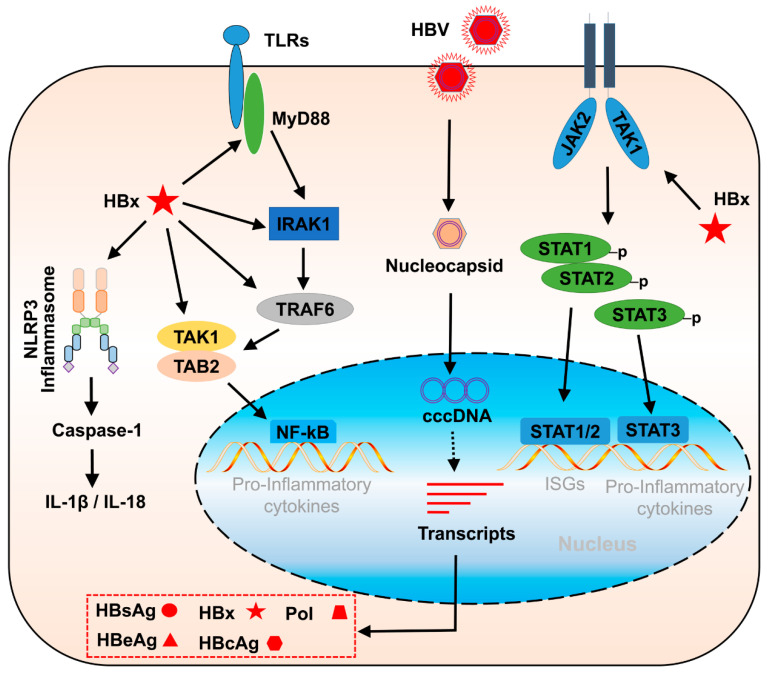
The promoting roles of Hepatitis B virus x protein (HBx) in the signal transduction of innate immunity and inflammation in HBV infection. HBx can activate TLR and nuclear factor-kappaB (NF-κB) signal pathways which promotes the expression of pro-inflammatory cytokines. HBx interacts with Janus Kinase 1 (JAK1), promoting the signal transducer and activator of transcription 1/2 (STAT1/2)-mediated ISG expression and STAT3-mediated cytokine expression. HBx promotes NLRP3 inflammasome activation to accelerate the release of IL-1β and IL-18.

**Figure 2 viruses-14-02275-f002:**
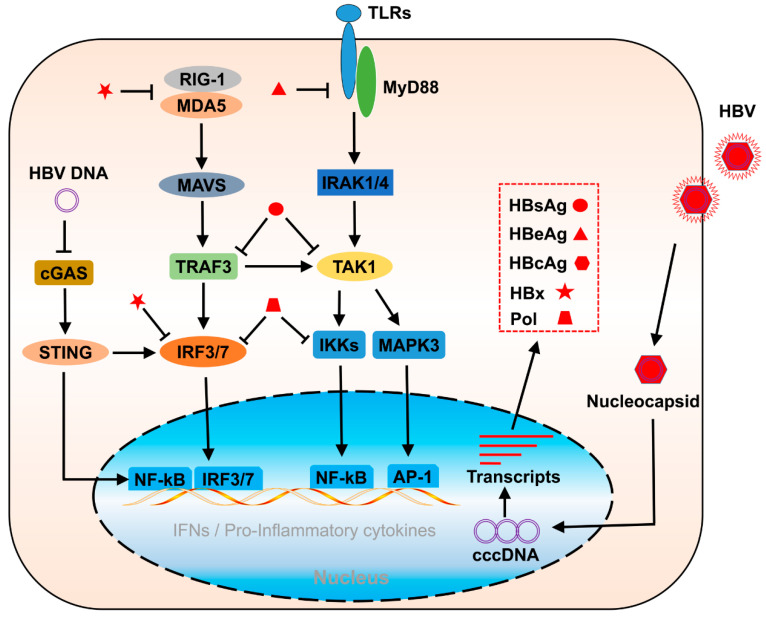
Schematic diagram of the inhibition of HBV protein in innate immunity and inflammation. HBx impairs IFN activation by inhibiting RIG-I and IRF3/7. HBeAg suppresses TLR-mediated innate immune response. HBsAg targets TRAF3 and TAK1, and HBV Pol targets IRF3/7 and IKKs, both of which lead to suppressive effects on the induction of ISGs and pro-inflammatory cytokines during HBV infection.

## Data Availability

Not applicable.

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
