# Peer review of "Innate Immunity, Inflammation, and Intervention in HBV Infection"

_viruses, 2022, doi:10.3390/v14102275_

Round 1
Reviewer 1 Report (Previous Reviewer 2)
In this manuscript, the authors summarized the interplay between HBV and innate immune responses, inflammation as well as the current therapeutic strategies for HBV infection though modulating innate immunity or inflammation. As a resubmitted manuscript, it is well modified now. Therefore, the reviewer recommends this for publication in Viruses after minor revisions.
1. Please make sure the clear description of the experimental conditions when citing the corresponding conclusions, such as transfection or infection, cell type or other models. It may help explain the contradictions between HBV and innate immunity.
2. Please search the whole text for some minor errors to be corrected.
For example,
"N6- m6A" in line 183, no space.
"TLR3,4" in line 314, "TLR-7" in line 321, please unify.
"IFN-λ1" in line 371, "IFN-lambda1" in line 376, please unify.
"by through" in line 383, delete one of them.
Author Response
Thanks for your comments. We have revised the manuscript carefully following your suggestions.
The Page numbers mentioned in the Response file are based on the status of “Track Changes” with all changes shown

Reviewer 2 Report (Previous Reviewer 3)
This paper mainly discussed the relationship between HBV infections and the immunological pathways. HBV is causing lots of trouble worldwide. The significance knowing the relationship between HBV infection and human immunity against viral infection is super important. The paper is very rich in content. However, there are still some big and small points that needs to be re-considered.
1) For the overall content: You didn’t mention anything about the HBV vaccines. Please discuss something related, such as the importance of having HBV vaccines? How protective are the vaccines? Why people are still in danger after receiving HBV vaccines? How many people are still in trouble now? Mentioning something like these will improve the significance of your study.
2) While you are talking about symptoms, please also mention how HBV is transmitted.
3) Rc-DNA (relaxed circular) and ccc-DNA (covalently closed circular), please put the full name first, and then use abbreviations. Also, please briefly explains what they are, since those are specific definitions for this type of viruses. This will help building some connections between the old texts and the new texts. The current transition feels wired. To solve this problem, you may simply want to make the new texts in a separated paragraph. In this way, you can add more details and definitions.
4) Line 63, please highlight the importance of treating early-stage HBV infections, such as any symptoms and sequela. This will help you emphasize the significance of the works.
5) You first mention HBx in figure 1 legends. This thing shows up as an abbreviation. Please mention what is HBx (Hepatitis B x protein) in the texts, and discuss its functions.
6) Line 71-81, when you summarize the inflammatory response caused by viral infections, please also mention what’s the eventual results for cells, like the cell fate after up-regulation of cytokines. Also, HBV infection could only induce pro-inflammatory cytokines to a very limited extent, so, from a large scale of perspectives, please re-mention the reactions/symptoms that patients may have at this situation.
7) For nuclear factor-kappa B, try to introduce it before line 148 since the abbreviations already appeared in the earlier sections. For IKKα, β and γ, try to mention what they are, at least explain their functions in brackets like what you did in line 152. In all of these immunology papers, it is always hard to track since there are so many abbreviations. Please go through the abbreviations and try to make things clear to readers.
8) Around line 166, the explanation of rc- and ccc- DNA, go back to comment 1). Overall, the flow of the paper is wired, since the new addings in yellow talking about HBV infection cycles showed up in both first paragraph and around line 170.
9) Any clinical data showing the current status of any drugs/therapeutic strategies? For instance, are there any known side effects?
10) On page 10 you have listed all the abbreviations. Please double check in your texts and make sure the full names are mentioned first. In addition, since there are so many abbreviations in the field of immunology, when you mention their full names, try to explain briefly on what they are, which will help readers getting through the contents.
Author Response
Thanks for your comments. We have revised the manuscript carefully following your suggestions.
The Page numbers mentioned in the Response file are based on the status of “Track Changes” with all changes shown.

This manuscript is a resubmission of an earlier submission. The following is a list of the peer review reports and author responses from that submission.
Round 1
Reviewer 1 Report
Review on the viruses-1779305
General points;
The authors described the importance of innate immunity, inflammation and then intervention in HBV infection. Basically, however, the description is just to list up the date reported so far. And actually, readers will not understand which/what is the most important. I would like to say that almost all data were obtained from transient transfection experiments but not infection experiment. Even though there were control experiments, such data would not reflect real infection status and would be overemphasized/ overestimated. The authors should reevaluate the importance by comparing the other viral infection model. How much does HBV influence innate immunity, inflammation or how much is HBV influenced by such? What is the best viral infection model that affect innate immunity and/or inflammation or is affected by such?
Thus, the authors should pick up the most reliable data and discuss the application for HBV infection. Even though experiment showed the importance of some molecules working in innate immunity and/or inflammation, such would not work as a therapeutic target (Expert Opin. Drug Discov. (2012) 7(7):597-611).
I would like to reiterate to describe what molecules were the most important in the HBV infection life cycle among innate immunity, inflammation and were promising as therapeutic targets.
Specific points;
1, English is not so bad but there are some grammatical errors, which should be corrected.
2. Estimation of 350 million of HBV infected patients worldwide might be overestimated. Please, check the newest information.
Reviewer 2 Report
The manuscript presented by Yang et al. describes the interplay between HBV and innate immune pathways and current therapeutic strategies for HBV infection via targeting innate immunity. It can be a valuable contribution to the field. However, several points should be improved as described below.
1. A part of HBV virology (such as genome, structure, life cycle) should be introduced before part 2 in this manuscript in order to support follow-up part.
2. In addition to TLR, RLR is another innate immune receptor to be activated by HBV. However, the authors did not discuss it in part 2.
3. As a review, it is not enough to simply list other people's work. Proper debate is required for contradictions between HBV and innate immunity. For example, the authors described the activation and inhibition of TLR pathway by HBV in this manuscript. Please try to explain.
4. Any schematic diagrams are welcome in this manuscript.
5. other issues:
Hepatocellular carcinoma should be replaced with hepatocellular carcinoma in abstract.
3.2kb should be replaced with 3.2 kb in introduction.
double stranded RNA should be included as the ligand of TLR in part 2.1.
viral proteins should be replaced with Viral proteins in conclusion.
Jak/Stat should be replaced with JAK-STAT in conclusion.
Reviewer 3 Report
This review article talked about how HBV infection mediates human innate immune system responses and inflammation including how immune responses are inactivated, different receptors, proteins that are involved in different pathways, the mechanisms behind immune results and what has been found or developed based on the relationship between HBV and innate immunity to stop HBV infections. The topic of this paper has strong significance to the field, since both virology and immunology strongly impact human life, and it is important to know the bridges between the two. Overall, the paper is rich in content and well-organized, just a few places need citations, like last sentence of the first paragraph 2.1., the introduction part of section 3. where you talk about the details of viral infection. It is recommended to make a citation after each solid statement in a sentence. Also, please refer figure 1 and figure 2 in your main text. There are so many abbreviations in an immunological pathway, and it is very easy to get lost. Telling the readers that there’s a figure that elucidates things in a more clear and direct way at a correct time point is super important.